# Whole-Transcriptome Sequencing Reveals a ceRNA Regulatory Network Associated with the Process of Periodic Albinism under Low Temperature in Baiye No. 1 (*Camellia sinensis*)

**DOI:** 10.3390/ijms24087162

**Published:** 2023-04-12

**Authors:** Cunbin Xu, Jinling Li, Hualei Wang, Huijuan Liu, Zhihai Yu, Zhi Zhao

**Affiliations:** 1College of Life Sciences, Guizhou University, Guiyang 550025, China; 2Guizhou Key Laboratory of Propagation and Cultivation on Medicinal Plants, Guizhou University, Guiyang 550025, China; 3College of Food and Pharmaceutical Engineering, Guizhou Institute of Technology, Guiyang 550003, China

**Keywords:** *Camellia sinensis*, Baiye No. 1, periodic albinism, whole transcriptome, ceRNA, non-coding RNA

## Abstract

The young shoots of the tea plant Baiye No. 1 display an albino phenotype in the early spring under low environmental temperatures, and the leaves re-green like those of common tea cultivars during the warm season. Periodic albinism is precisely regulated by a complex gene network that leads to metabolic differences and enhances the nutritional value of tea leaves. Here, we identified messenger RNAs (mRNAs), long noncoding RNAs (lncRNAs), circular RNAs (circRNAs), and microRNAs (miRNAs) to construct competing endogenous RNA (ceRNA) regulatory networks. We performed whole-transcriptome sequencing of 12 samples from four periods (Bud, leaves not expanded; Alb, albino leaves; Med, re-greening leaves; and Gre, green leaves) and identified a total of 6325 differentially expressed mRNAs (DEmRNAs), 667 differentially expressed miRNAs (DEmiRNAs), 1702 differentially expressed lncRNAs (DElncRNAs), and 122 differentially expressed circRNAs (DEcircRNAs). Furthermore, we constructed ceRNA networks on the basis of co-differential expression analyses which comprised 112, 35, 38, and 15 DEmRNAs, DEmiRNAs, DElncRNAs, and DEcircRNAs, respectively. Based on the regulatory networks, we identified important genes and their interactions with lncRNAs, circRNAs, and miRNAs during periodic albinism, including the ceRNA regulatory network centered on miR5021x, the *GAMYB*-miR159-lncRNA regulatory network, and the *NAC035*-miR319x-circRNA regulatory network. These regulatory networks might be involved in the response to cold stress, photosynthesis, chlorophyll synthesis, amino acid synthesis, and flavonoid accumulation. Our findings provide novel insights into ceRNA regulatory mechanisms involved in Baiye No. 1 during periodic albinism and will aid future studies of the molecular mechanisms underlying albinism mutants.

## 1. Introduction

Albino tea mutants are important germplasm seed resources of high economic value and they can be classified into temperature-sensitive, light-sensitive, and ecologically insensitive albino variants [1]. Baiye No. 1 (which is often referred to as Anji Baicha or White Leaf No. 1) is a low-temperature-sensitive, periodic albino variant that has albino young shoots in the early spring when the environmental temperature is below 20–22 °C. The leaves re-green like those of common tea cultivars when the temperature rises above 22 °C in late spring. In most crops, albinism results in yield loss or death, but tea products made from albino shoots have higher nutritional value, more umami, a less bitter taste, and a higher market value than normal varieties.

The unique phenotypic and physiological characteristics of Baiye No. 1 have motivated studies of its underlying molecular mechanisms. Multi-omics studies have been conducted on Baiye No. 1 and other types of albino tea trees [1]. At the transcriptional level, different patterns of gene expression have been observed during periodical albinism through cDNA microarrays [2], amplified fragment length polymorphisms [3], quantitative real-time PCR (qRT-PCR) [4,5], and RNA sequencing (RNA-seq) [6]. Differences in protein expression have been studied using two-dimensional electrophoresis [7], the tandem mass tag isobaric labeling technique and acetyl-proteome profiling [8], and succinyl-proteome profiling [9] at the post-translational level. Several studies have conducted omics analyses of other albino or yellow species, such as Xiao-yue-ya [10,11,12,13], Yu-jin-xiang [14], Huang-jin-ya [15,16], Hua-bai-1 [17], Bai-ji-guan [18], and a novel albino branch of tea plants [19].

Whole-transcriptome analysis employs high-throughput sequencing technology to characterize associations among messenger RNAs (mRNAs), microRNAs (miRNAs), long non-coding RNAs (lncRNAs), and circular RNAs (circRNAs) and clarify the roles of non-coding RNAs (ncRNAs) in biological processes at the post-transcriptional level [20,21,22]. The competing endogenous RNA (ceRNA) hypothesis explains the relationship between coding RNAs and ncRNAs. The ceRNA hypothesis suggests that mRNAs and lncRNAs/circRNAs can interact through microRNA response elements and that lncRNAs/circRNAs act as miRNA sponges to absorb miRNAs, thereby inhibiting their silencing effects [23,24]. Lu et al. (2020) found that lncRNAs (lncR9A, lncR117, and lncR616) regulate the expression of their target genes, such as CSD1 and Dn1, through miR398 in winter wheat, which initiates the cold stress response [25]. Jing et al. (2022) identified ceRNA networks associated with drought tolerance in *Medicago truncatula* with *Rhizobium Symbiosis* [26]. Few studies have examined ceRNAs in tea plants, as well as the regulatory mechanism underlying the albino–green transformation in albino tea at the transcriptional and post-transcriptional levels. No studies to date have comprehensively identified and analyzed the miRNAs, lncRNAs, and circRNAs comprising ceRNA networks that are involved in the periodic albinism of Baiye No. 1.

Here, we performed a whole-transcriptome sequencing of Baiye No. 1 tea plants during periodic albinism. The differential expression of mRNAs, lncRNAs, circRNAs, and miRNAs was characterized at four stages (Bud, leaves not expanded; Alb, albino leaves; Med, re-greening leaves; and Gre, green leaves), and the regulatory roles of these RNAs were clarified using the Gene Ontology (GO) and Kyoto Encyclopedia of Genes and Genomes (KEGG) databases. Additionally, a ceRNA network was established to clarify the regulatory mechanism underlying the periodic albinism of Baiye No. 1 as well as changes in the contents of metabolites.

## 2. Results

### 2.1. Phenotypic Characteristics of Baiye No. 1

In early spring, the dormant or burgeoning buds of Baiye No. 1 are yellow-green (Figure 1A, Bud), bleaching with the shoots sprouting under temperatures below 20 °C (Figure 1A, Alb). With increasing temperatures, new leaves gradually turn green (Figure 1A, Med and Gre). The main compound contents in fresh tea leaves of Baiye No. 1 at four stages of periodic albinism (Bud, Alb, Med, and Gre) were analyzed, including photosynthetic pigments (Figure 1B), total free amino acids, and tea polyphenols (Figure 1C). There were significant differences in chlorophyll and carotenoid contents in the four stages, among which the contents in the Alb stage were the lowest, and the contents in the Gre stage were the highest. From the chlorophyll a/b ratio, the ratio in the Alb stages was the lowest (1.51), and there were no significant differences between the other three phases (Figure 1D). Additionally, the total free amino acids in the Alb stage were the highest, with the order Alb > Med > Bud > Gre, and the total free amino acids in the Alb stage were 3.25 times those in the Gre stage. The content of total tea polyphenols was opposite to that of total free amino acids in the order of Gre > Bud > Med > Alb, and the content of tea polyphenols in Gre was 2.02 times that in Alb. The phenol–ammonia ratio is an important index in evaluation of the taste of tea. A phenol–ammonia ratio < 6 is generally believed to be suitable for making green tea. Gre has the highest phenol–ammonia ratio (5.38), with the order Gre > Bud > Med > Alb, and the phenol–ammonia ratio at the Alb stage is 0.82 (fresh weight) (Figure 1D).

### 2.2. Global Response of RNAseq

RNA-seq was performed on 12 tea buds or leaves from four periods (Bud, Alb, Med, and Gre) (Figure 1A); three biological replicates were performed for each sample using the Illumina HiSeqTM 4000 platform. A total of 190 Gb raw data comprising 1,269,858,132 raw reads were generated by RNA-seq after adapters and low-quality reads were removed; a total of 1,265,834,922 (99.68%) clean reads were generated (Appendix A). The transcriptome sequencing data were deposited with NCBI under the BioProject number PRJNA927052. The clean reads were aligned with the ribosomal RNA (rRNA), and mapped reads were removed. The remaining 1,227,738,338 (96.68%) reads were aligned to the tea plant reference genome (*Camellia sinensis var. sinensis* ‘Shuchazao’) [27], and a total of 955,679,733 (75.26%) reads were mapped to the reference genome. The numbers of reads mapped to exons, introns, and intergenic regions are shown in Figure 2A, and the high-quality clean reads were used to identify mRNAs, lncRNAs, and circRNAs.

### 2.3. Identification of Differentially Expressed mRNAs (DEmRNAs)

We identified 5715 novel genes (compared with the ‘Shuchazao’ genome), using Stringtie to reconstruct the transcripts. This, combined with the 30,173 annotated genes, revealed a total of 35,888 genes, including 27,929 genes detected in these transcript libraries; the fragments per kilobase of transcript per million mapped reads (FPKMs) of all genes are shown in Appendix A. Principal component analysis (PCA) revealed that all the samples were clustered along the principal component 1 (PC1) axis (Figure 2B), but Bud1 was far from Bud2 and Bud3 along the principal component 2 axis. In addition, the three Med samples were clustered with Gre samples, which was caused by the small differences in their phenotypes and the fact that these samples were derived from adjacent periods of development. Finally, most samples had high loadings on PC1 (94.2%), and the Pearson correlation coefficients of the Bud1 vs. Bud2 and Bud1 vs. Bud3 comparison groups were 0.8113 and 0.874, respectively; both being >0.8, we used all samples for the next analysis.

A total of 6325 DEmRNAs were identified by group comparisons (Appendix A), and the numbers of DEmRNAs in each comparison group are shown in Figure 2C. The largest difference was observed in the Bud vs. Gre comparison group, with 4538 and 914 genes with up-regulated and down-regulated expression, respectively, in green leaves, followed by the Bud vs. Alb comparison group (2013 up-regulated and 530 down-regulated) and the Alb vs. Gre comparison group (472 up-regulated and 412 down-regulated). We detected DEmRNAs in the Bud vs. Med comparison group (524 up-regulated and 92 down-regulated), and the number of DEmRNAs was three times and 8.8 times lower in the Bud vs. Med comparison group than in the Bud vs. Alb and Bud vs. Gre comparison groups, respectively. The numbers of DEmRNAs were substantially lower in the Alb vs. Med comparison group (37 up-regulated and 24 down-regulated) and the Med vs. Gre comparison group (13 up-regulated and 15 down-regulated) than in the other comparison groups, suggesting that re-greening might involve a slow accumulation process. The short Time-series Expression Miner software [28] was used for the gene expression trend analysis, for which 26 expression patterns were set, and the results were as follows (Figure 2D). In the expression trend analysis, there were 11 profiles that were significantly enriched, including profiles 21, 4, 15, 24, 3, 1, 22, 2, 16, 0, and 5 (Figure 2D). The profiles could be divided into two types: the first was that of DEmRNAs down-regulated in bleaching or partial bleaching periods (Bud, Alb, and Med), including the significantly enriched profiles 15, 16, 21, 22, 24, and 25. The second type was that of high expression in the bleaching process and low expression in the greening process, including profiles 0, 1, 2, 3, and 4. Profile 21 was the most-enriched pattern, with 3969 DEmRNAs; these genes were lowly expressed in the Bud stage, then highly expressed in Alb and maintained in the Gre stage, indicating that the silencing or down-regulation of the DEmRNAs in the bud stage contributed to the leaves bleaching.

GO and KEGG enrichment analyses were performed to explore the potential functions of the DEmRNAs (Appendix A, Figure 2E,G). GO enrichment analysis revealed 416 significantly enriched GO terms (*p* < 0.05) (Appendix A), and GO terms for each comparison group are shown in Appendix A. Most of the DEmRNAs were annotated to the following GO terms in the biological process (BP) category: reproduction, activation of MAPK activity, cytokinesis, carbohydrate metabolic process, phototransduction, and flavonoid metabolic process. In the cellular component (CC) category, most of the DEmRNAs were annotated to the following GO terms: condensed chromosome, peroxisome, chloroplast, photosystem, intrinsic component of membrane, membrane part, and thylakoid part. In the molecular function (MF) category, most of the DEmRNAs were annotated to the following GO terms: nucleic acid binding, succinate-CoA ligase activity, water transmembrane transporter activity, drug binding, beta-glucosidase activity, and IAA-amino acid conjugate hydrolase activity. According to the KEGG enrichment analysis, these DEmRNAs were significantly enriched in 27 KEGG metabolic pathways (*p* < 0.05) (Figure 2G, Appendix A), and enriched pathways for each comparison group are shown in Appendix A. The enriched pathways included glycine, serine, and threonine metabolism; lysine biosynthesis; flavonoid biosynthesis; starch and sucrose metabolism; photosynthesis; photosynthesis—antenna proteins; thiamine metabolism; porphyrin and chlorophyll metabolism; autophagy—other eukaryotes; and phagosome. Therefore, the enrichment analysis showed that these DEmRNAs might contribute to the albinism, high amino acid content, and low polyphenol content of Baiye No. 1.

We further examined the Alb vs. Gre comparison group (Figure 2F) because the samples in these two stages had the largest phenotypic differences, including major differences in color, polyphenol and chlorophyll content, and free amino acid content (Figure 1). DEmRNAs in this comparison group were significantly enriched in pathways such as photosynthesis—antenna proteins, photosynthesis, flavonoid biosynthesis, and porphyrin and chlorophyll metabolism (Appendix A). Some of these pathways were also enriched in other comparison groups, such as photosynthesis—antenna proteins and photosynthesis in the Bud vs. Gre comparison group (Appendix A) and photosynthesis in the Bud vs. Alb comparison group (Appendix A). The DEmRNAs in these pathways were associated with phenotypic differences. We focused on the DEmRNAs in the Alb vs. Gre comparison group (Appendix A); eight DEmRNAs were enriched in photosynthesis—antenna proteins, including *TEA_001579 (CAB7)*, *TEA_001699 (CAB21)*, *TEA_007351 (LHCA4)*, *TEA_008676 (CAB40)*, *TEA_015416 (CAB13)*, *TEA_017260 (CAB40)*, *TEA_020597 (LHCB5)*, and *TEA_027142 (CAB40)*, and the expression of these genes was significantly down-regulated during the Alb stage. Eleven DEmRNAs were enriched in photosynthesis, including *TEA_001265 (petA)*, *TEA_001277 (petA)*, *TEA_001460 (psbA)*, *TEA_002319 (psaB)*, *TEA_012480 (PSBP1)*, *TEA_014408 (atpA)*, *TEA_019916 (PSAH)*, *TEA_024135 (petA)*, *MSTRG.145455 (atpI)*, *MSTRG.148586 (petB)*, and *MSTRG.159981 (psb28)*, and the expression of most of these genes was down-regulated during the Alb stage, with the exception of *TEA_001460 (psbA)*. Five DEmRNAs were enriched in porphyrin and chlorophyll metabolism, including *TEA_004733 (COX15)*, *TEA_013230 (PORA)*, *TEA_013354 (CLH1)*, *TEA_019887 (HEMC)*, and *TEA_026668 (PORA)*, and the expression of these was down-regulated during the Alb stage, with the exception of *TEA_004733 (COX15)*. Nine DEmRNAs were enriched in flavonoid biosynthesis, including *TEA_005966 (FLS)*, *TEA_006385 (ACT)*, *TEA_011819 (ACT-2)*, *TEA_011933 (CYP75A1)*, *TEA_014811 (FL)*, *TEA_020913 (CHS3)*, *TEA_029911 (ACT)*, *MSTRG.83458 (ACT)*, and *MSTRG.86901 (FLS)*, and the expression of these genes was down-regulated during the Alb stage, with the exception of TEA_014811 (FL). The differential expression of these genes might be responsible for the periodic albinism of Baiye No. 1 and the changes in the contents of metabolites.

### 2.4. Identification of Differentially Expressed lncRNAs (DElncRNAs)

We identified 12,037 lncRNAs (Appendix A, Figure 3A), and these could be divided into five categories; the numbers of each type are shown in Figure 3B. The number of intergenic lncRNAs was the highest (62.35%). The results of the principal component analysis (PCA) of lncRNAs were similar to those for mRNAs and revealed that all the samples were clustered along the principal component 1 (PC1) axis (Figure 3C), with the three Med samples clustered with the Gre and Alb samples. A total of 1702 DElncRNAs were detected, and they are listed in Appendix A. The numbers of DElncRNAs in each comparison group are shown in Figure 3D. The greatest numbers of DElncRNAs were observed in the Bud vs. Gre comparison group (1362 up-regulated and 144 down-regulated) and the Bud vs. Alb comparison group (603 up-regulated and 53 down-regulated); only one and five DElncRNAs were detected in the Med vs. Gre and Alb vs. Med comparison groups, respectively. In the expression trend analysis, seven profiles were significantly enriched, including profiles 21, 3, 1, 24, 15, 4, and 0 (Figure 3F); three profiles were down-regulated in bleaching or partial bleaching periods (Bud, Alb, and Med); and four profiles showed high expression in the bleaching process.

lncRNAs can cis-regulate the expression of adjacent upstream and downstream genes. A total of 5026 cis-lncRNAs were predicted, and these were associated with 4958 mRNAs, forming 6747 lncRNA-mRNA pairs (Appendix A). A total of 191 cis-DElncRNAs were associated with 183 mRNAs, forming 203 DElncRNA-mRNA pairs (Appendix A). GO and KEGG enrichment analyses were performed to explore the potential functions of the cis-lncRNA and cis-DElncRNA target genes (Appendix A). A total of 130 significantly enriched GO terms were identified (*p* < 0.05) (Figure 3E, Appendix A), including glucose metabolic process, vitamin metabolic process, heme biosynthetic process, small molecule metabolic process, arginine metabolic process, and fatty acid metabolic process in the BP category; intracellular, cell, cell part, plastid thylakoid, and intracellular part in the CC category; and SNARE binding, ferrochelatase activity, protein transmembrane transporter activity, protein transporter activity, macromolecule transmembrane transporter activity, macromolecule transmembrane transporter activity, and amino acid transmembrane transporter activity in the MF category. KEGG analysis was performed, and a total of 12 significantly enriched KEGG metabolic pathways were identified (*p* < 0.05) (Figure 3G, Appendix A), including valine, leucine and isoleucine degradation, carbon fixation in photosynthetic organisms, SNARE interactions in vesicular transport, carbon metabolism, steroid biosynthesis, vitamin B6 metabolism, porphyrin and chlorophyll metabolism, purine metabolism, pyrimidine metabolism, DNA replication, spliceosome, and phagosome. We identified lncRNAs, such as *MSTRG.120018.4*, which regulates the expression of *TEA_003046 (SIRB)*, and *MSTRG.87254.1*, which regulates the expression of *TEA_015525 (OVA3)*, that are significantly enriched in porphyrin and chlorophyll metabolism (Appendix A).

### 2.5. Identification of Differentially Expressed circRNAs (DEcircRNAs)

A total of 8853 circRNAs were identified from the RNA libraries by find_circ software [29] (Appendix A). The numbers of different types of circRNA sources are shown in Figure 4A; there were a total of six types of circRNAs sources, and the most common were antisense circRNAs (31.14%), followed by one_exon circRNAs (21.59%). The lengths of circRNAs ranged from 1 to 2000 bp (Figure 4B), and most circRNAs were 201 to 300 bp in length (31.75%), followed by ones that were 101 to 200 bp in length (28.63%). A total of 122 DEcircRNAs were identified by group comparisons (Appendix A), and the numbers of DEcircRNAs in each group comparison are shown in Figure 4C. In the expression trend analysis, only three profiles were significantly enriched—profiles 21, 14, and 15 (Figure 4E); the DEcircRNAs of these were down-regulated in bleaching or partial bleaching periods.

To further clarify the functions of DEcircRNAs, GO and KEGG analyses were performed on the DEcircRNA-source genes (Appendix A). GO enrichment analysis revealed 277 significantly enriched GO terms (*p* < 0.05) (Figure 4D, Appendix A), and GO terms for each comparison group are shown in Appendix A. DEcircRNAs were enriched in sulfur amino acid metabolic process; plastid organization; photosynthesis; photosynthesis, light reaction; photosynthetic electron transport chain; serine family amino acid metabolic process; porphyrin-containing compound metabolic process; glycine metabolic process; and starch metabolic process in the BP category. In the CC category, DEcircRNAs were enriched in cytoskeleton; microtubule-associated complex; plastid envelope; plastid; microtubule cytoskeleton; proton-transporting two-sector ATPase complex; plastid thylakoid; proton-transporting two-sector ATPase complex, catalytic domain; and non-membrane-bounded organelle. In the MF category, DEcircRNAs were enriched in adenine nucleotide transmembrane transporter activity; transcription factor activity, protein binding; core DNA-dependent RNA polymerase binding promoter specificity activity; motor activity; endopeptidase activity; and hydrogen ion transmembrane transporter activity. According to the KEGG enrichment analysis, these DEcircRNAs were significantly enriched in eight KEGG pathways (*p* < 0.05) (Figure 4F, Appendix A), and the enriched pathways for each comparison group are shown in Appendix A. DEcircRNA-source genes were enriched in the following KEGG pathways: phenylpropanoid biosynthesis, photosynthesis, RNA degradation, metabolic pathways, ABC transporters, cyanoamino acid metabolism, ribosome biogenesis in eukaryotes, and phagosome.

### 2.6. Identification of Differentially Expressed miRNAs (DEmiRNAs)

To fully understand the miRNA repertoire during periodic albinism, miRNA sequencing was performed on 12 tea buds or tea leaves from four stages (Bud, Alb, Med, and Gre); three biological replicates were performed for each sample, and the samples were sequenced using an Illumina HiSeqTM 2500 sequencing platform. The raw data were deposited with the National Center for Biotechnology Information (NCBI) under the BioProject number PRJNA927052. A total of 165,042,781 clean reads were obtained after adapters and low-quality reads were removed; a total of 128,972,506 (78.14%) high-quality clean tags were generated for the next step of the analysis. The data for each sample are shown in Appendix A. We calculated the length distribution of all clean tags, and the results are shown in Figure 5A. Two peaks were observed at 21 nucleotides (nt) and 24 nt, and the number of Taq sequences 24 nt in length was the largest (22.75%), which is consistent with the typical characteristics of Dicer enzyme cutting products. Finally, a total of 3644 miRNAs (959 known and 2685 novel miRNAs) were identified (Appendix A). We identified 667 DEmiRNAs via group comparisons (Appendix A); the numbers of DEmiRNAs in each comparison group are shown in Figure 5B. We also predicted 3926 target genes for 667 DEmiRNAs in each group comparison (Appendix A). In the expression trend analysis, six profiles were significantly enriched—profiles 7, 4, 6, 21, 3, and 1 (Figure 5D); unlike previous DEmRNAs (Figure 2C), DElncRNAs (Figure 3D), and DEcircRNAs (Figure 4C), most of the DEmiRNAs were up-regulated in bleaching or partial bleaching periods (Bud, Alb, and Med), such as 7, 4, 6, 3, and 1, which suggests that miRNA may be related to down-regulated mRNA expression during periodic albinism.

GO and KEGG analyses were performed on the target genes of DEmiRNAs (Appendix A) to clarify their functions. GO enrichment analysis revealed 367 significantly enriched GO terms (*p* < 0.05) (Figure 5C, Appendix A), and GO terms for each comparison group are shown in Appendix A. DEmiRNAs were enriched in 272 GO terms in the BP category, including regulation of transcription, DNA-templated, transcription, and protein phosphorylation; 32 GO terms in the CC category, including nucleus, plasma membrane, and chloroplast envelope; and 63 GO terms in the MF category, including ATP binding, DNA binding, and DNA binding transcription factor activity. We also performed KEGG enrichment analysis on the target genes of the DEmiRNAs, and 23 significantly enriched KEGG metabolic pathways were identified (*p* < 0.05) (Figure 5E, Appendix A). The enriched pathways for each comparison group are shown in Appendix A. The significantly enriched pathways included alanine, aspartate and glutamate metabolism, tryptophan metabolism, monobactam biosynthesis, indole alkaloid biosynthesis, phenylpropanoid biosynthesis, pentose phosphate pathway, fructose and mannose metabolism, photosynthesis, plant–pathogen interaction, metabolic pathways, biosynthesis of secondary metabolites, glycosaminoglycan degradation, glycosphingolipid biosynthesis—globo and isoglobo series, steroid biosynthesis, glycerophospholipid metabolism, thiamine metabolism, folate biosynthesis, porphyrin and chlorophyll metabolism, taurine and hypotaurine metabolism, selenocompound metabolism, zeatin biosynthesis, phosphatidylinositol signaling system, and plant hormone signal transduction. DEmiRNAs were also enriched in photosynthesis, chlorophyll synthesis, porphyrin and chlorophyll metabolism, and amino acid metabolism, which contribute to albinism and changes in the contents of metabolites in Baiye No. 1.

### 2.7. ceRNA-miRNA-Target Gene Regulatory Network

To clarify the global regulatory network of protein-coding RNAs and ncRNAs that are involved in albinism, ceRNA networks were constructed based on the ceRNA theory using DEmRNAs, DEmiRNAs, DElncRNAs, and DEcircRNAs. A total of 6325 DEmRNAs, 667 DEmiRNAs, 1702 DElncRNAs, and 122 DEcircRNAs were identified as differentially expressed in ‘Baiye No. 1’ during periodic albinism. As ceRNAs are regulated by miRNAs, we predicted the target genes of DEmiRNAs as a first step. ceRNA analysis revealed 35 DEmiRNAs, 38 DElncRNAs, 15 DEcircRNAs, and 112 DEmRNAs (Appendix A). Cytoscape software (https://cytoscape.org (accessed on 12 April 2020)) was used to visualize the regulatory relationships (Figure 6).

miRNAs link mRNAs and lncRNAs/circRNAs by inhibiting mRNA expression and being regulated by lncRNAs/circRNAs. According to the mRNA-miRNA-lncRNA/circRNA regulatory relationships, the ceRNA regulatory network relationships can be classified into three categories. In the first category, miR5021x was the main node, and circRNAs (*novel-circ-004921* and *novel-circ-002272*) and lncRNAs (*MSTRG.123813.1*) were intermediate nodes that formed a complex regulatory network with more miRNAs, mRNAs, and lncRNAs (Figure 6A). In the second category, mRNAs, via multiple miRNAs and interactions with multiple lncRNAs, such as *MSTRG.80659* and *MSTRG.158018*, can interact with six miRNAs, and these miRNAs can interact with six lncRNAs (Figure 6B). In the third category, the expression of single or multiple mRNAs can be regulated using one or two miRNAs as response elements, along with lncRNAs or circRNAs (Figure 6C).

GO and KEGG enrichment analyses were performed on the ceRNA regulatory network (the functional annotations for ncRNAs were determined according to the functions of their related mRNAs), and the results are shown in Figure 7. We identified terms and pathways related to the chloroplast, including thylakoid, plastid thylakoid, photosynthesis, and terpenoid backbone biosynthesis; protein and amino acid metabolism, including valine, leucine, and isoleucine degradation, histidine metabolism, and peptidase activity; plant stress response, including plant–pathogen interaction, glutathione metabolism, phenylpropanoid biosynthesis, plant hormone signal transduction, homeostatic process, plant-type cell wall organization or biogenesis, and kinase activity; and nucleic acid and protein modification, including base conversion or substitution editing, protein serine/threonine kinase activity, exopeptidase activity, and transferring activity.

According to the functional descriptions of the genes, we identified various DEmRNAs that were directly or indirectly involved in albinism, chlorophyll metabolism, flavonoid synthesis, amino acid metabolism, and response to low-temperature stress. A total of 69 genes were regulated by miR5021x, such as *MSTRG.159981 (psb28)*, which is involved in photosynthesis. miR1879y was connected to miR5021x through *novell-circ-004921*, and 11 genes were regulated by miR1879y, including *TEA_018352 (WRKY53)*, *TEA_020929 (Carnmt1)*, *TEA_014326 (VPS41)*, *TEA_015784 (AAP7)*, *TEA_003146 (DG1)*, *TEA_024920* (Adenosine/AMP deaminase domain-containing protein), *TEA_030061 (GSTU25)*, *TEA_007691 (HSP82)*, *TEA_022251 (SYP43)*, *TEA_003522 (PUX10)*, and *TEA_011010 (RUK)*. Seven genes were regulated by miR3630x, including *TEA_028449 (8-Aug)*, *TEA_003416 (RLP1)*, *TEA_001558 (TMN12)*, *TEA_003940 (LIL3.2)*, *TEA_026541 (GH3.17)*, *TEA_027326 (GNS1)*, and *TEA_015619 (RLP1)*. *TEA_028449 (8-Aug)* was predicted to encode AUGMIN subunit 8, which is required for chloroplast development in *Arabidopsis*. *TEA_003940 (LIL3.2)* was predicted to encode light-harvesting complex-like protein 3 isotype 1, which is involved in photosynthesis—antenna proteins, and reductions in LIL3 might affect pigment-protein assembly and chlorophyll synthesis, result in a chlorotic phenotype, and impair photosynthetic performance [30,31,32]. *TEA_026541 (GH3.17)* encodes indole-3-acetic acid-amido synthetase, and the expression of this gene is down-regulated in response to cold in *Arabidopsis* [33]. The expression of *TEA_022757 (GAM1/GAMYB)* and *TEA_018956 (SPEAR2)* was regulated by miR159y, and the expression of *TEA_007521 (NAC035)* was regulated by miR319x. GAMYB and NAC035 are two transcription factors that play important roles in life activities. Our results revealed that the mRNA-miRNA-lncRNA/circRNA regulatory network might be involved in periodic albinism. Additional studies are needed to clarify the functions of ceRNAs, miRNAs, and target genes to elucidate the mechanisms underlying periodic albinism.

### 2.8. Verification of the RNA-seq Results

To confirm the quality of the RNA-seq data and the expression patterns of miRNAs, lncRNAs, circRNAs, and mRNAs during the periodic albinism of Baiye No. 1, qRT-PCR analyses were conducted on 10 DEmRNAs, 6 DElncRNAs, 6 DEcircRNAs, and 6 DEmiRNAs; the results are shown in Figure 8. The expression of these genes inferred from the RNA-seq and qRT-PCR analyses was similar, suggesting that the RNA-seq data were reliable. However, trace expression levels of some genes (*TEA_022757*, *TEA_027671*, *TEA_003146*, *MSTRG.145201.1*, and *novel_circ_002159*) were detected in the qRT-PCR analyses, and no expression of these genes was detected according to the RNA-seq data.

## 3. Discussion

High-throughput sequencing data enhance our understanding of the functions of different types of RNAs, especially ncRNAs, that play important roles in life activities [22]. In the present work, we performed biochemical testing and whole-transcriptome sequencing on four periods (Bud, Alb, Med, and Gre) during periodic albinism of Baiye No. 1 tea plants. The biochemical analysis showed that the total free amino acid content was highest during the Alb period, while the contents of chlorophyll, carotenoid, chlorophyll a/b, polyphenols, and total polyphenols to total free amino acids (P/A) were lowest during the Alb period. These factors contribute to making the tea taste more umami, less bitter, and more nutritious.

We obtained whole-transcriptome information on Baiye No. 1 tea plants during four stages of periodic albinism. As a result, we identified a total of 6325 DEmRNAs, 667 DEmiRNAs, 1702 DElncRNAs, and 122 DEcircRNAs in Baiye No. 1 during periodic albinism. The DEmRNA expression trend analysis showed that most genes were significantly down-regulated in the Bud and Alb periods, which is consistent with previous research [3,6]. However, our interesting finding is that the largest number of DEmRNAs were significantly down-regulated in the Bud period. Initially, we thought that the morphology of the leaves may cause a large number of differential genes. However, we found that the number of DEmRNAs in Bud vs. Med was significantly lower than that in Bud vs. Gre (8.8-fold) and Bud vs. Alb (3-fold), indicating that developmental morphology might not be the main cause of the large differences in gene expression. Furthermore, we conjectured that the response of Baiye No. 1 to low temperatures had already begun in the Bud stage. The numbers of DEmRNAs in Alb vs. Med and Med vs. Gre were much lower than in Alb vs. Gre, which may suggest that the re-greening of albino leaves may gradually restore normal expression levels, which is a new discovery. Functional enrichment analysis of DEmRNAs between comparison groups or different trend patterns pinpointed that the DEmRNAs were enriched in such pathways as photosynthesis—antenna proteins, photosynthesis, flavonoid biosynthesis, and porphyrin and chlorophyll metabolism, and so on. These pathways may be involved in periodic albinism and the content changes in chlorophyll, amino acids, and polyphenols; earlier studies have also confirmed that these pathways may play a role [1,6].

We detected many lncRNAs during the periodic albinism which have lengths over 200 bp and do not encode proteins but play important roles in regulating various biological processes [34]. lncRNAs can cis-regulate the expression of adjacent upstream and downstream genes, such as *MSTRG.120018.4*, which regulates the expression of *TEA_003046 (SIRB)*, and *MSTRG.87254.1*, which regulates the expression of *TEA_015525 (OVA3)*. The expression of TEA_003046 (SIRB) was up-regulated in the Bud stage, and SIRB promotes the conversion of uroporphyrinogen III to siroheme [35,36], which may promote heme synthesis. TEA_015525 (OVA3) encodes glutamate-tRNA ligase, which catalyzes the first step of chlorophyll synthesis, and the expression of this gene was down-regulated in the Bud and Alb stages. The differential expression of SIRB and OVA3 may lead to a decrease in chlorophyll content and cause albinism.

We also discovered many circRNAs and miRNAs during the periodic albinism. To better understand their functions, we conducted a competing endogenous RNA (ceRNA) analysis. The analysis revealed 35 DEmiRNAs, 38 DElncRNAs, 15 DEcircRNAs, and 112 DEmRNAs, and this information was used to establish a ceRNA network (Figure 6). Based on the ceRNA regulatory network relationships, we identified 35 DEmiRNAs including seven known miRNAs (miR5021x, miR3630x, miR1879y, miR319x, miR159y, miR5502y, and miR1871y) and 28 newly predicted miRNAs. Notably, miR319 [37], miR159 [38,39], miR5021 [40], miR3630x [41], and miR1871y [42] play a role in the response of plants to low temperature, drought, high salt, and other stresses, and they affect the metabolism of chlorophyll, flavonoids, amino acids, and other secondary metabolites. Moreover, a functional enrichment analysis was carried out on the differentially expressed mRNAs (DEmRNAs) participating in the ceRNA network. The outcome revealed a significant association of numerous genes with the metabolism of pigments, flavonoids, and amino acids, suggesting the potential involvement of the ceRNA regulatory network in periodic albinism.

In a network diagram drawn in Cytoscape (Figure 6), we found that most genes form a complex regulatory network centered around miR5021x. miR5021x can regulate 68 target genes. Functional analysis showed that these genes are enriched in multiple pathways, such as photosynthesis, chlorophyll synthesis, and amino acid synthesis. miR5021x is complementary to six circRNAs (*novel-circ-005739*, *novel-circ-003483*, *novel-circ-001540*, *novel-circ-007581*, *novel-circ-004921*, and *novel-circ-002272*) and four lncRNAs (*MSTRG.69243.1*, *MSTRG.121872.1*, *MSTRG.107960.1*, and *MSTRG.123813.1*) in the ceRNA network. miR5021x also forms complex regulatory networks with more miRNAs, mRNAs, and lncRNAs via circRNAs (*novel-circ-004921* and *novel-circ-002272*) and lncRNAs (*MSTRG.123813.1*) (Figure 6A). miR5021 is involved in several processes related to the regulation of plant growth, development, and stress resistance. For example, the expression of miR5021 alters the activity of zinc finger protein, which affects the response to salt stress in *Paulownia fortunei* [43]. In our study, both *TEA_027671* and *TEA_009103* encoded zinc finger proteins. The expression of miR5021 is up-regulated in peach [40] and watermelon [44] in response to cold stress, indicating that it plays a role in the cold stress response. In addition, some circRNAs act on miR5021 family members in poplar under low nitrogen stress [45]. We also identified a possible RNA competition mechanism that regulates photosynthesis involving *MSTRG.159981 (psb28)* -miR5021x-lncRNA/circRNA. The expression of *MSTRG.159981 (psb28)* was significantly down-regulated during the Alb stage, and *psb28* encodes an important component of PSII; it thus plays an important role in maintaining the stability of PSII, including its assembly and repair [46,47]. The down-regulation of *psb28* in the Bud and Alb stages not only affects the assembly of photosystem II (PSII) but also affects chlorophyll synthesis [46,47,48]. Previous studies have shown that the absence of Psb28 can significantly increase contents of chlorophyll synthesis precursors, such as protoporphyrin IX and Mg-Protoporphyin, but inhibit chlorophyll synthesis in *Arabidopsis* [48]. The content of Mg-Protoporphyin IX increases significantly during the Alb stage in Baiye No. 1 [49], but the content of chlorophyllide significantly decreases during the Alb stage; this pattern is similar to previous observations of the *psb28* null mutant of *Arabidopsis* [48]. In the network diagram constructed by miR5021, miR1879y was connected to miR5021x through novell-circ-004921 and regulated 11 genes. Among them, DG1 is a pentratricopeptide repeat protein involved in regulating early chloroplast development and chloroplast gene expression in *Arabidopsis*, and young, inner leaves of dg1 mutants are initially very pale but gradually green; mature outer leaves appear similar to those of wild-type plants [50,51], which accords with our observations of Baiye No. 1.

We also identified the *GAMYB*-miR159y-lncRNA regulatory network. miR159y targets *TEA_018956 (SPEAR2)* and *TEA_022757 (GAM1/GAMYB)* and is regulated by two lncRNAs (*MSTRG.145201.1* and *MSTRG.81550.2*) (Figure 6C). miR159-*GAMYB* is a common regulatory pathway in plants; *GAMYB* encodes the transcription factor R2R3 MYB, which inhibits plant growth and development; miR159 can silence the expression of *GAMYB* and induce normal plant growth [52]. However, the high expression of miR159 results in dwarf tobacco plants and inhibits responses to pathogens [53]; the mechanism underlying the role of the miR159-*GAMYB* pathway in mediating the resistance of plants to abiotic stress has been previously clarified [52]. Li et al. (2021) analyzed the correlation between miRNAs and the synthesis of secondary metabolites in different tissues of tea trees and verified that the miRNA-*GAMYB* pathway occurs in tea trees [54]. The up-regulated expression of miR159 inhibits the activity of *GAMYB*. The MYB transcription factor can synergistically regulate the response of plants to cold stress by interacting with a bHLH transcription factor, and this bHLH transcription factor can activate the expression of *CBF* and *DFR* genes and positively regulate the response of plants to low-temperature stress and flavonoid accumulation [38].

The NAC035-miR319x-circRNA regulatory network might increase the sensitivity of Baiye No. 1 to low temperatures. The expression of miR319x was significantly up-regulated in Baiye No. 1 in the Bud stage, but previous studies suggest that miR319 expression is down-regulated in rice [55], sugarcane [56], Hemerocallis [57], and other plants under cold stress. In the hardy tea variety Yin-shuang, the expression of miR319 is significantly down-regulated under cold stress [37], and the expression of gma-miR319n is up-regulated 12 h after cold treatment according to degradation group sequencing of Baiye No. 1; these findings are consistent with the results of our study. miR319x targets *TEA_007521 (NAC035)*, and the NAC035 transcription factor can improve the cold tolerance of plants by positively regulating the expression of *COR15A* and *KIN1*. The knockout of *NAC035* increases the sensitivity of plants to cold stress [58]. The high expression of miR319x in Baiye No. 1 at the Bud stage inhibited the expression of NAC035, which increased the sensitivity of tea trees to cold stress. miR319x can also be regulated by lncRNAs (e.g., *MSTRG.121609.1*) and circRNAs (e.g., *novel-circ-004923*).

However, the PCA analysis of the relationship between the samples showed that the three Med samples were mixed with Alb and Gre samples, which was caused by the small differences in their phenotypes, making it difficult to select relatively uniform samples. Therefore, a precise method is needed in future studies, such as single-cell transcriptomics or spatial transcriptomics, etc.

In sum, our results provide new insights into the regulatory network of periodic albinism in Baiye No. 1 by shedding light on new regulatory ceRNA (mRNA-miRNA-lncRNA/circRNA) relationships. As we have shown, the ceRNA networks participate in photosynthesis, porphyrin and chlorophyll synthesis, flavonoid synthesis, amino acid metabolism, and response to low-temperature stimulation, which is basically consistent with the regulation mechanism predicted in this and previous works. All of the lncRNAs, circRNAs, miRNAs, target genes, and their interactions identified provide a useful resource that will aid future research. Additional experimental studies and computational analyses will be needed to clarify the biological processes underlying periodic albinism induced by low temperatures in tea mutants.

## 4. Materials and Methods

### 4.1. Plant Materials

Field observations were made and samples were collected from March 20 to April 20, 2019, at the Ya’shang Tea Planting Base (longitude: E 107°21′33″, latitude: N 28°18′33″), Zheng’an County, Guizhou Province, China. The tea plants were propagated using a clonal method (cuttings) and planted in 2010, and they had a stable phenotype every year. Buds and leaf samples of Baiye No. 1 (*Camellia sinensis*) with different colors at the four stages were collected in this experiment (Figure 1A). Bud-stage samples were pre-albino buds nearly 1.5 cm long; Alb-stage samples were albino leaves; Med leaves were leaves that had begun to re-green around the main veins; and Gre leaves were completely green. The leaves were placed into liquid nitrogen immediately after they were collected and stored in a refrigerator in a laboratory at −80 °C.

### 4.2. Measurements of Main Metabolites

The chlorophyll, total polyphenol, and total amino acid contents were determined by spectrophotometry (HITACHI, Tokyo, Japan). The contents of chlorophyll were extracted based on previously reported methods [59], using 95% alcohol, and were determined at 665, 649, and 470 nm. The total polyphenols were extracted by 70% methanol and determined by 10% Folin–Ciocalteu, and gallic acid was used as a standard solution. Total amino acids were extracted by 75% alcohol and determined by 2% ninhydrin, and glutamic acid was used as a standard solution.

### 4.3. RNA Library Construction and Sequencing

Total RNA was extracted from 12 samples using Trizol reagent (Invitrogen, Carlsbad, CA, USA). RNA quality was assessed on an Agilent 2100 Bioanalyzer (Agilent Technologies, Palo Alto, CA, USA). After total RNA was extracted, rRNAs were removed to retain mRNAs and ncRNAs using a Hieff NGS^®^ MaxUp rRNA Depletion Kit (Plant). The enriched mRNAs and ncRNAs were fragmented into short fragments using a fragmentation buffer and reverse-transcribed into cDNA using random primers. Second-strand cDNA was synthesized using DNA polymerase I, RNase H, dNTP (dUTP instead of dTTP), and buffer. Next, the cDNA fragments were purified using a QiaQuick PCR extraction kit (Qiagen, Venlo, The Netherlands) and end-repaired; poly(A) tails were then added and ligated to Illumina sequencing adapters. Uracil-N-Glycosylase was used to digest the second-strand cDNA. The digested products were size-selected by agarose gel electrophoresis, PCR-amplified, and sequenced using the Illumina HiSeqTM 4000 system by the Gene Denovo Biotechnology Co. (Guangzhou, China).

Small RNAs ranging from 18 to 30 nt were enriched by polyacrylamide gel electrophoresis after total RNA was extracted. The 3′ adapters were then added, and 36–44 nt RNAs were enriched. The 5′ adapters were then ligated to the RNAs. The ligation products were reverse-transcribed by PCR amplification, and PCR products from 140 to 160 bp were enriched to generate a cDNA library; they were then sequenced using Illumina HiSeqTM 2500 by the Gene Denovo Biotechnology Co. (Guangzhou, China).

### 4.4. Data Filtering and mRNA, lncRNA, and circRNA Identification

Raw mRNA, lncRNA, and circRNA data were filtered using fastp [60] (version 0.18.0); reads containing adapters, unknown nucleotides, and low-quality bases were removed. Next, we used Bowtie2 [61] (version 2.2.8) to remove the rRNA-mapped reads, and the clean reads were mapped to the reference genome (*Camellia sinensis var. sinensis* ‘Shuchazao’, GCA_004153795.2) using HISAT2 [62]. The transcripts were reconstructed using Stringtie [63,64] (version 1.3.4), and HISAT2 was used to identify novel transcripts. After the final transcriptome was generated, transcript abundances were quantified using StringTie according to FPKM values. Differential expression analysis of mRNAs between pairs of groups was performed using DESeq2 [65], and edgeR [66] was used to analyze differential expression between pairs of samples. Genes/transcripts with the parameters false discovery rate (FDR) < 0.05 and |log2(fold change (FC)| > 1 were considered DEmRNAs.

CNCI [67] (version 2) and CPC [68] (version 0.9-r2) (http://cpc.cbi.pku.edu.cn/ (accessed on 2 September 2019)) were used to identify lncRNAs with default parameters; the lncRNAs retained were potential non-protein-coding RNAs identified by both programs. lncRNA-mRNA association analysis was performed, and the interactions between lncRNAs and mRNAs were classified into antisense, cis-, and trans-regulatory mechanisms. The software RNAplex [69] (version 0.2) (http://www.tbi.univie.ac.at/RNA/RNAplex.1.html (accessed on 2 September 2019)) was used to predict correlations between antisense-lncRNA and mRNA. lncRNAs less than 100 kb upstream or downstream of a target gene were classified as cis-regulators. Trans-lncRNAs were identified according to correlations (Pearson correlation coefficient ≥ 0.999) of expression between lncRNAs and protein-coding genes. The transcript abundances of lncRNAs were determined according to FPKM values, and transcripts with FDR < 0.05 and |log2(FC)| > 1 were considered DElncRNAs.

circRNAs were identified by find_circ [29] through circRNA splicing, which was determined by anchor reads that aligned in the reverse orientation (head-to-tail). The anchor alignments were then extended such that the complete read aligned, and the breakpoints were flanked by GU/AG splice sites. A candidate circRNA was called if it was supported by at least two unique back-spliced reads in at least one sample. The type, chromosome distribution, and length distribution of identified circRNAs were analyzed. To quantify circRNAs, back-spliced junction reads were scaled to reads per million mapped reads. The edgeR package was used to identify significant DEcircRNAs according to the following criteria: |log2(FC)| > 1 and *p* < 0.05.

### 4.5. miRNA Identification and Target Gene Prediction

The raw reads obtained from the sequencing machines were filtered by removing low-quality reads and adapters. All of the clean tags were then searched against the miRbase database [70] (Release 22) to identify known miRNAs, and the novel miRNA candidates were identified using the software Mireap_v0.2 [71] from the unannotated tags. The miRNA expression levels were calculated and normalized to transcripts per million, and significant DEmiRNAs were identified according to the following criteria: |log2(FC)| > 1 and *p* < 0.05. The software Patmatch [72] (version 1.2) was used to predict target genes.

### 4.6. The PCA Analysis and Gene Expression Pattern Analysis

Based on the expression results of known genes in each sample, principal component analysis (PCA) and Pearson correlation coefficients between samples were used to understand the repeatability of samples and to help exclude outlier samples. PCA was performed using the R package gmodels (http://www.rproject.org (accessed on 20 September 2019)) method, and Pearson’s correlation coefficient was used to evaluate the repeatability between samples. The closer the correlation coefficient was to 1, the better the repeatability of the two parallel experiments, and 0.8–1.0 was considered highly correlated.

Gene expression pattern analysis is used to cluster genes with similar expression patterns. To examine the expression pattern of DEGs, the expression data for each sample (in the order of treatment) were normalized to 0, log2 (v1/v0), and log2(v2/v0) and were then clustered by Short Time-series Expression Miner software (STEM) [28].

### 4.7. Construction of the ceRNA Network

We identified significant DEmRNAs and DElncRNAs using the criteria |log2(FC)| > 1 and FDR < 0.05 and DEcircRNAs and DEmiRNAs using the criteria |log2(FC)| > 1 and *p* < 0.05. The software Patmatch was used to predict miRNA target genes and lncRNA/circRNA pairs. Next, we conducted a two-step screening. Pairs with Spearman rank correlation coefficients < −0.3 were negatively co-expressed ceRNA-miRNA pairs, and pairs with Pearson correlation coefficients > 0.5 were co-expressed lncRNA-mRNA and circRNA-mRNA pairs; these were used to obtain candidate ceRNA relationships. Cytoscape software (https://cytoscape.org (accessed on 12 April 2020)) was used to visualize regulatory relationships.

### 4.8. GO and KEGG Enrichment Analyses

GO and KEGG enrichment analyses were performed on DEmRNAs, lncRNA target genes, circRNA source genes, miRNA target genes, and genes in the ceRNA network. The genes were mapped to GO terms in the GO database (http://www.geneontology.org/ (accessed on 20 April 2020)), the numbers of genes enriched for each term were determined, and significantly enriched GO terms relative to the genome background were identified using a hypergeometric test (*p* < 0.05). KEGG pathway analysis was performed using the KEGG pathway database (www.kegg.jp/kegg/kegg1.html (accessed on 20 April 2020)), and significantly enriched metabolic pathways or signal transduction pathways relative to the whole-genome background were identified using a hypergeometric test (*p* < 0.05).

### 4.9. qRT-PCR

The expression levels of selected DEmRNAs, DElncRNAs, DEcircRNAs, and DEmiRNAs were validated using qRT-PCR. Total RNAs and small RNAs were extracted using an RNAprep pure Plant Kit (Tiangen, Beijing, China) as per the manufacturer’s instructions. Reverse transcription was performed using the PrimeScript RT reagent Kit (Takara, Dalian, China) and SYBR Premix Ex Taq Kit (Takara, Dalian, China), and qRT-PCR was performed following the manufacturer’s instructions. mRNAs and lncRNAs were reverse-transcribed using oligo (dT) primers and random 6-mer primers, respectively. The miRNAs were reverse-transcribed using the downstream primers of the U6 endogenous reference gene and the specific stem-loop primers. For reverse transcription of the circRNA, we used the qRT-PCR downstream primers that we designed near the back-spliced site and the random 6-mer primers provided in the kit. GAPDH was used as an internal control for mRNAs, lncRNAs, and circRNAs. U6 was used as an endogenous control for the miRNAs. All qRT-PCR reactions were performed in three technical replicates and three biological replicates. The primer sequences and procedures are listed in Appendix A. The qRT-PCR was incubated in a CFX Connect™ Real-Time System (Bio-Rad, Hercules, CA, USA) with the following conditions: 95 °C for 2 min; followed by 40 cycles at 95 °C for 15 s, 60 °C for 30 s; and finally at 95 °C for 15 s, 65 °C for 30 min, and 95 °C for 15 s. The relative expression was then analyzed via the 2−∆∆CT method.

Statistical analysis was performed using Microsoft Excel and IBM SPSS Statistics 24. One-way analysis of variance was used to evaluate the significance of differences among groups, and the threshold for statistical significance was *p* < 0.05.

## 5. Conclusions

Whole-transcriptome sequencing was performed on 12 buds or leaves from four stages (Bud, Alb, Med, and Gre) of periodic albinism in *Camellia sinensis* cv. Baiye No. 1. We constructed a ceRNA regulatory network which comprised 35 DEmiRNAs, 38 DElncRNAs, 15 DEcircRNAs, and 112 DEmRNAs, and identified several regulatory networks involved in the response to cold stress, photosynthesis, chlorophyll synthesis, amino acid synthesis, and flavonoid accumulation. Firstly, we identified the regulatory network centered on miR5021x; the functional analysis showed enrichment in multiple pathways, such as photosynthesis, chlorophyll synthesis, and amino acid synthesis, and genes such as *MSTRG.159981 (psb28)* were down-regulated in Alb, which may not only affect the stability of PSII but also reduce chlorophyll synthesis. Then, in the *GAMYB*-miR159-lncRNA regulatory network, the expression of miR159 was up-regulated, and this inhibited the activity of MYB transcription factors *(TEA_022757)*. MYB transcription factors can synergistically regulate the response of plants to cold stress by interacting with bHLH transcription factors, and this also has positive regulatory effects on the process of flavonoid accumulation. Furthermore, in the *NAC035*-miR319x-circRNA regulatory network, the expression of miR319x was significantly up-regulated during Alb due to low-temperature stimulation, and the expression of miR319x was down-regulated in other plants, as well as cold-resistant tea trees. miR319x inhibited the expression of *TEA_007521 (NAC035)*, which increased the sensitivity of Baiye No. 1 to cold stress. Our findings revealed mRNAs and ncRNAs involved in periodic albinism in Baiye No. 1 and will aid future studies aimed at exploring the molecular mechanisms underlying the regulation of periodic albinism in tea plants.

## Figures and Tables

**Figure 1 ijms-24-07162-f001:**
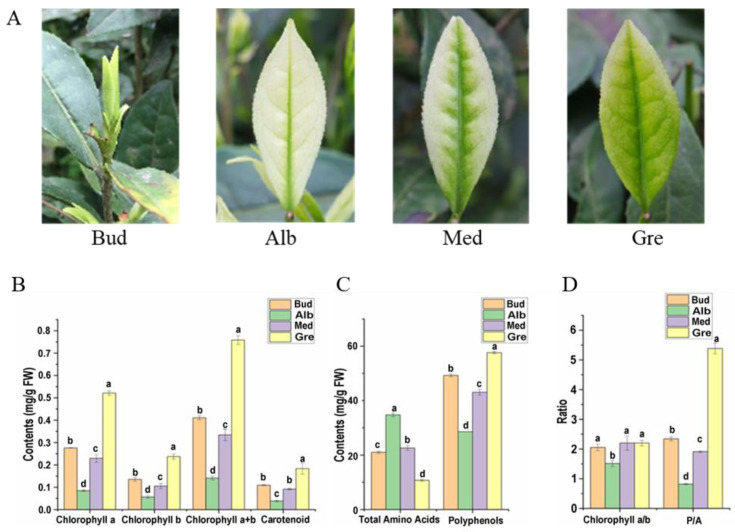
Phenotypes and main compound contents of Baiye No. 1. (**A**) The re-greening process of the leaf in tea cultivar Baiye No. 1. Bud, leaves not expanded; Alb, albino leaves; Med, re-greening leaves; Gre, green leaves. (**B**) Changes in pigment contents in four periods. (**C**) Changes in total free amino acids and total polyphenol contents in four periods. (**D**) The ratio of chlorophyll a to chlorophyll b (chlorophyll a/b) and total polyphenols to total free amino acids (P/A). The data are expressed as mean values ± standard deviations, and different lowercase letters indicate significant differences (*p* < 0.05) as determined by ANOVA.

**Figure 2 ijms-24-07162-f002:**
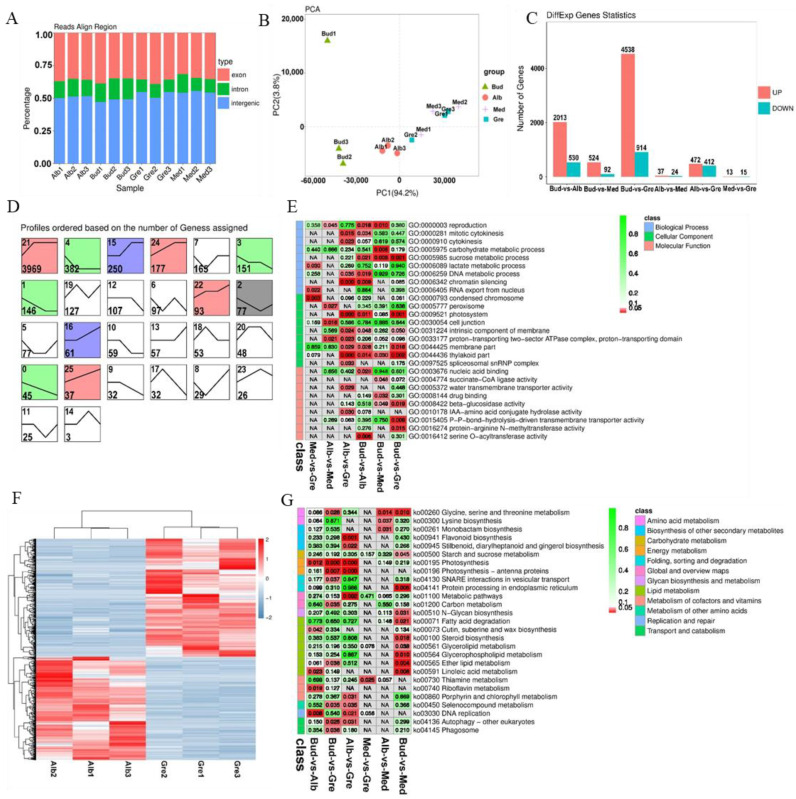
Identification and analysis of mRNAs in Baiye No. 1. (**A**) The alignment of all mapped reads was used to calculate the distribution of reads in the reference genome. The regions aligned to the genome were classified as exonic, intronic, or intergenic. (**B**) PCA of the mRNA expression patterns in samples. (**C**) Statistics of DEmRNAs among different groups. (**D**) Expression trend analysis of DEmRNAs, listed by the number of genes. The number in the upper left corner is the profile number, and the number in the lower left corner is the number of genes; the colors show significantly enriched profiles (*p* < 0.05). (**E**) Partial significance GO terms for the DEmRNAs of different comparison groups. See Appendix A for details. The coloration indicates the magnitudes of the significant differences, and the numbers represent *p*-values. Red indicates significant (*p* < 0.05) enrichment, green indicates enriched but not significant, and NA indicates not enriched. (**F**) KEGG pathway assignments for all the DEmRNAs of different comparison groups. Red indicates significant (*p* < 0.05) enrichment, green indicates enriched but not significant, the numbers represent *p*-values, and NA indicates not enriched. (**G**) The heatmap of DEmRNAs involved in Alb vs. Gre. The DEmRNAs were hierarchically clustered and mapped using the FPKM values. Colors indicate the normalized signal intensities, as defined in the bar: blue indicates low expression levels, and red indicates high expression levels.

**Figure 3 ijms-24-07162-f003:**
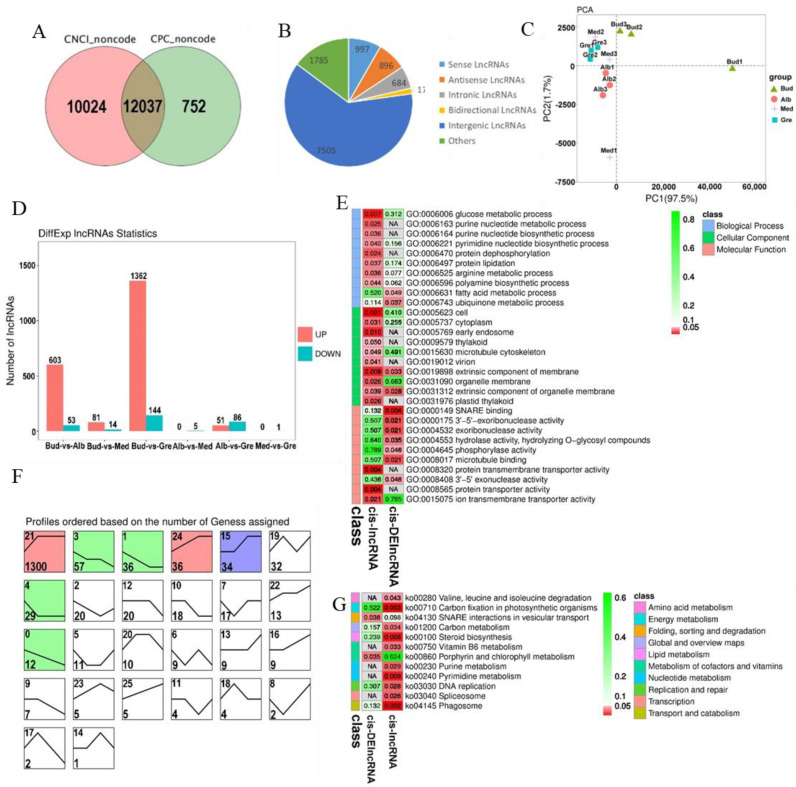
Identification and analysis of lncRNAs in Baiye No. 1. (**A**) Venn diagram of CNCI and CPC forecast results. (**B**) Numbers of different lncRNA types. (**C**) PCA of the lncRNA expression patterns in samples. (**D**) Statistics of DEcircRNAs among different groups. (**E**) Partially significant GO terms for cis-lncRNAs and cis-DElncRNAs. See Appendix A for details. The colors indicate the magnitudes of significant differences, and the numbers represent *p*-values. Red indicates significant (*p* < 0.05) enrichment, green indicates enriched but not significant, and NA indicates not enriched. (**F**) Expression trend analysis of DElncRNAs, listed by the number of genes. The number in the upper left corner is the profile number, and the number in the lower left corner is the number of genes; the colors show significantly enriched profiles (*p* < 0.05). (**G**) KEGG pathway assignments for cis-lncRNAs and cis-DElncRNAs. Red indicates significant (*p* < 0.05) enrichment, green indicates enriched but not significant, the numbers represent *p*-values, and NA indicates not enriched.

**Figure 4 ijms-24-07162-f004:**
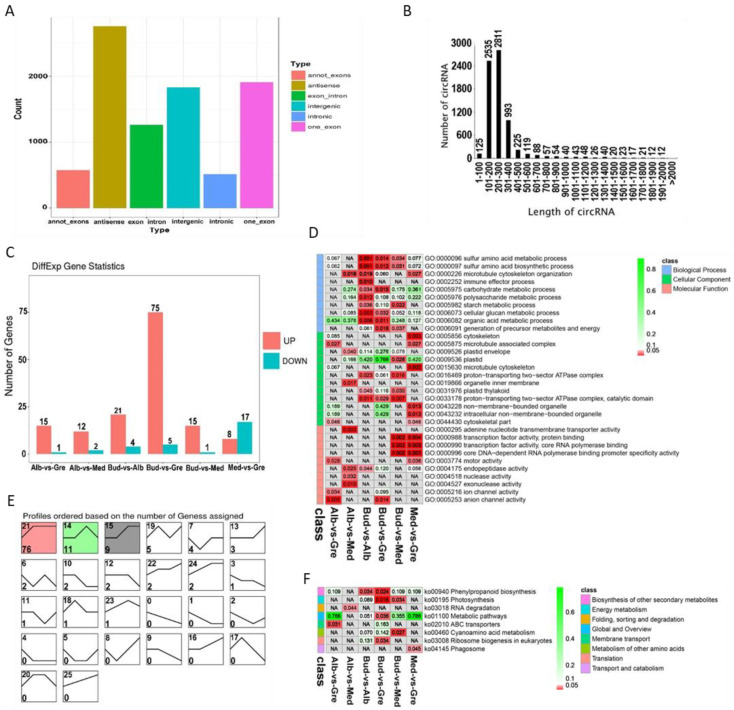
Identification and analysis of circRNAs in Baiye No. 1. (**A**) Distribution of different types of circRNAs. (**B**) Length distribution of all identified circRNAs. (**C**) Statistics of DEcircRNAs among different groups. (**D**) Partially significant GO terms for DEcircRNA-source genes of different comparison groups. See Appendix A for details. Colors indicate the magnitudes of significant differences, and the numbers represent *p*-values. Red indicates significant (*p* < 0.05) enrichment, green indicates enriched but not significant, and NA indicates not enriched. (**E**) Expression trend analysis of DEcircRNAs, listed by the number of genes. The number in the upper left corner is the profile number, and the number in the lower left corner is the number of genes; the colors show significantly enriched profiles (*p* < 0.05). (**F**) KEGG pathway assignments for DEcircRNA-source genes. Red indicates significantly (*p* < 0.05) enriched, green indicates enriched but not significant, the numbers represent *p*-values, and NA indicates not enriched.

**Figure 5 ijms-24-07162-f005:**
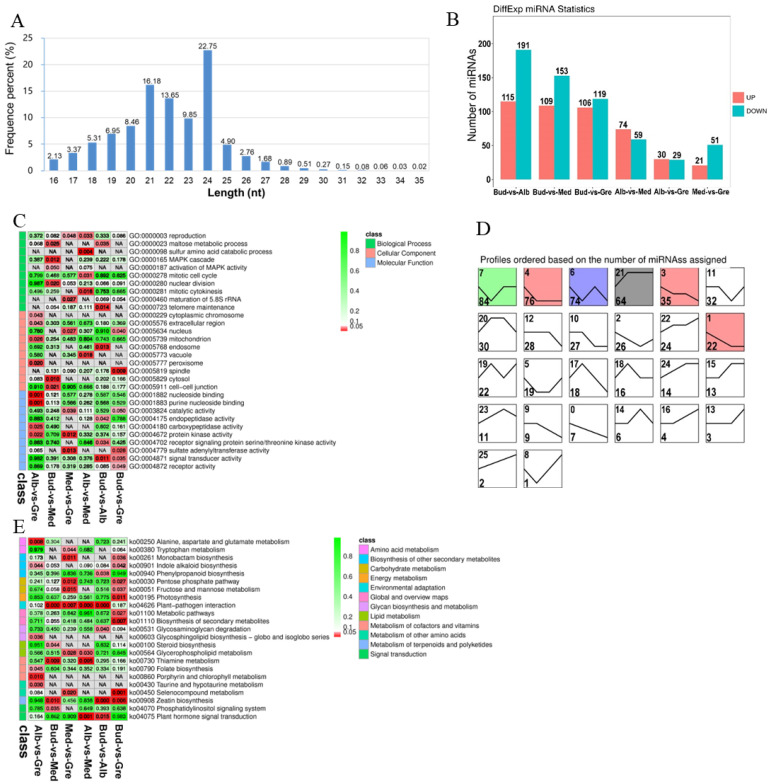
Identification and analysis of miRNAs in Baiye No. 1. (**A**) Length distribution of all identified miRNAs. (**B**) Statistics of DEmiRNAs among different groups. (**C**) Partially significant GO terms for DEmiRNA target genes of different comparison groups. See Appendix A for details. Colors indicate the magnitudes of significant differences. Red indicates significant (*p* < 0.05) enrichment, green indicates enriched but not significant, and NA indicates not enriched. (**D**) Expression trend analysis of DEmiRNAs, listed by the number of genes. The number in the upper left corner is the profile number, and the number in the lower left corner is the number of genes; the colors show significantly enriched profiles (*p* < 0.05). (**E**) KEGG pathway assignments for DEmiRNA target genes. Red indicates significant (*p* < 0.05) enrichment, green indicates enriched but not significant, the numbers represent *p*-values, and NA indicates not enriched.

**Figure 6 ijms-24-07162-f006:**
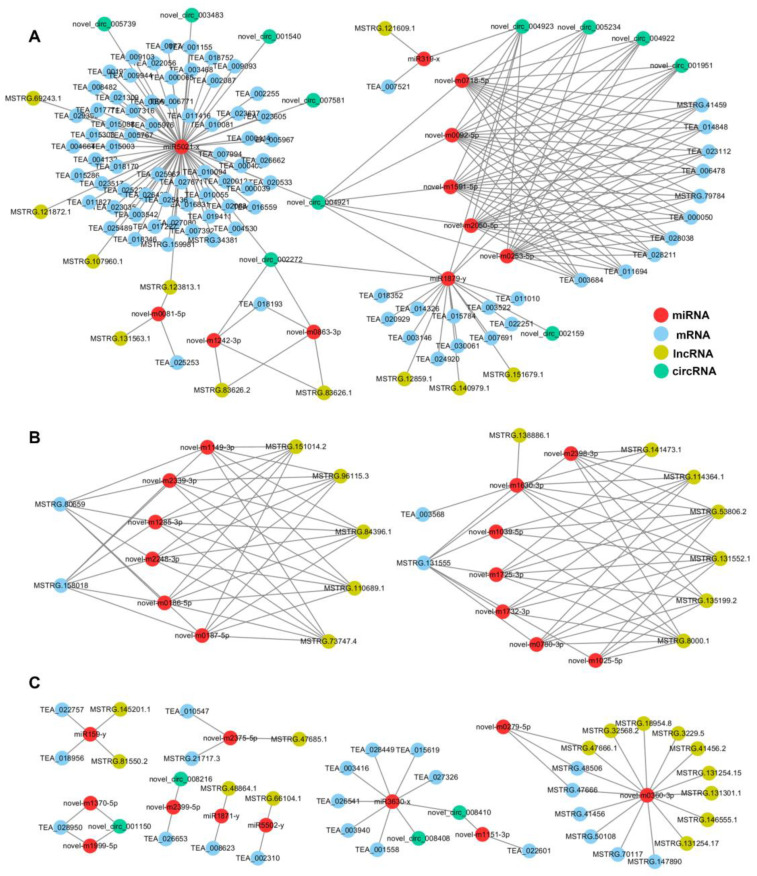
The ceRNA regulatory network. (**A**) A complex regulatory network comprising miR5021x as the main node, circRNAs (*novel-circ-004921* and *novel-circ-002272*) and lncRNAs (*MSTRG.123813.1*) as intermediate nodes, and other miRNAs, mRNAs, and lncRNAs. (**B**) Interactions of mRNAs with multiple lncRNAs are mediated by interactions with multiple miRNAs. For example, *MSTRG.80659* and *MSTRG.158018* can interact with six miRNAs, and these miRNAs can interact with six lncRNAs. (**C**) The expression of single or multiple mRNAs is regulated via single or two miRNAs as response elements, along with lncRNAs or circRNAs.

**Figure 7 ijms-24-07162-f007:**
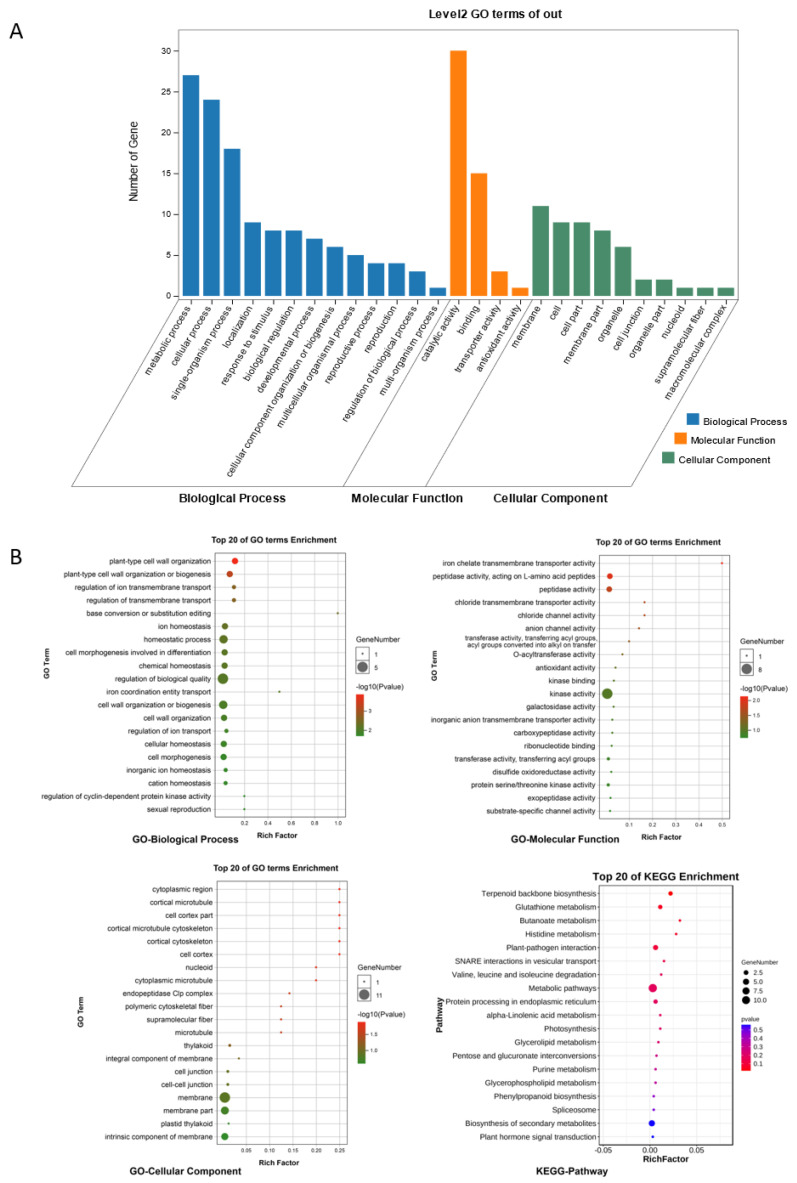
Results of GO and KEGG enrichment analyses of ceRNAs. (**A**) Histogram of Level 2 GO terms for ceRNA. (**B**) Top 20 GO terms and KEGG pathways for ceRNAs.

**Figure 8 ijms-24-07162-f008:**
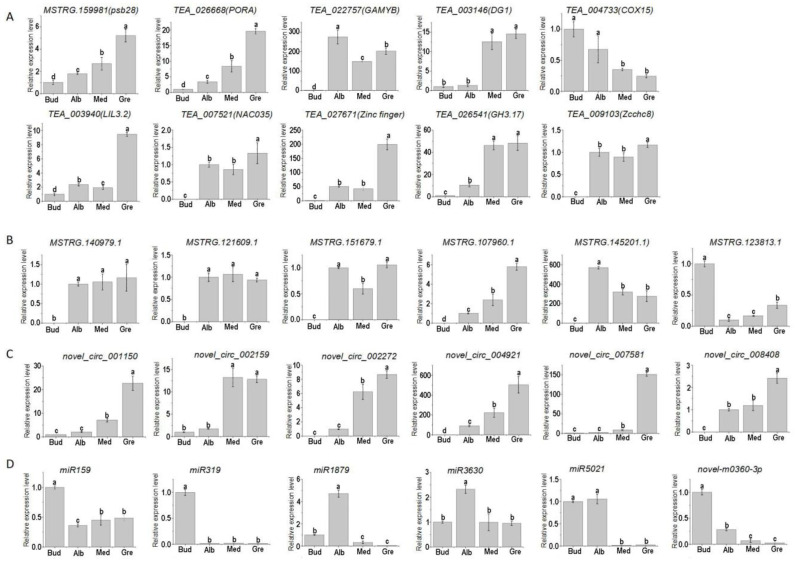
Gene expression of qPCR. Relative expression levels of 10 DEmRNAs (**A**), DElncRNAs (**B**), DEcircRNAs (**C**), and DEmiRNAs (**D**) involved in periodic albinism; the data are expressed as mean values ± standard deviations for three biological replicates based on qRT-PCR, and different lowercase letters indicate significant differences (*p* < 0.05) as determined by ANOVA.

## Data Availability

The RNA-seq raw data have been deposited with NCBI with the BioProject number PRJNA927052, and other data presented in this study are available from the corresponding author on reasonable request.

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
