# Peer review of "Whole-Transcriptome Sequencing Reveals a ceRNA Regulatory Network Associated with the Process of Periodic Albinism under Low Temperature in Baiye No. 1 (Camellia sinensis)"

_ijms, 2023, doi:10.3390/ijms24087162_

Round 1

Reviewer 1 Report

The presented manuscript is devoted to an interesting topic - temporary albinism in the tea bush. The experiment was carried out a big and big plus - this is a biochemical analysis. And although the methods of biochemical analysis used are already outdated, they greatly embellished the experiment based on computer programs. It should be noted that the authors did not limit themselves to determining the amount of activated RNAs, but also to determining the functional role in which they participate. However, based on your data, the effect of albinism causes a whole cascade of changes in the metabolism and biosynthesis of systems that are not even remotely related to this effect? How can you explain it. Further, does the tea bush have all albino leaves or only some of them? Have you looked at the spectra of chlorophylls in leaves of different periods of development? Under certain conditions, for example, under low illumination, the maxima of the spectra of chlorophylls a and b shift to shorter wavelengths, but their content remains the same as in green plants. Probably, the problem of temporary albinism can be associated with a slow maturation of chlorophylls at low temperatures. Different maturation rates can also be the reason for the difference in BUD1, BUD2, BUD3 repeats. Are these repeats? You have not provided a description of them. These comments do not detract from the value of this study and can be recommended for publication.

However, there are some remarks.

Figure 1 - the labels on the x-axis are gone.

117- autophagy =jther eukaryotes. What is this?

201 -Photosynthesis- and further in the test it is necessary to write with a small letter, as on line 176

238 - rephrase the sentence - "only one and five..."

543 - Italic gene names

Reviewer 2 Report

My comments can be found in the attached manuscript.

Round 2

Reviewer 2 Report

Thanks for making the suggested changes and I will be glad to see the MS in print.